# Equivariant CNNs via Hypernetworks

## Abstract

In geometric deep learning, numerous works have been dedicated to enhancing neural networks with the ability to preserve symmetries, a concept known as equivariance. Group Equivariant Convolutional Networks ($G$-CNNs) achieve rotation and reflection equivariance on Convolutional Neural Networks (CNNs). While showing a significant improvement when processing rotation-augmented datasets such as randomly rotated MNIST, training $G$-CNNs on a dataset with little rotational variation, such as regular MNIST, typically leads to a performance drop compared to a regular CNN. In this study, we first empirically observe the performance imbalance across different variation of MNIST in $G$-CNNs, and discuss how the $G$-CNN filters is a contributing factor to this imbalance. To avoid such imbalance, we propose a Hypernetwork-based Equivariant CNN (HE-CNN) to generate CNN filters that inherently exhibit rotational equivariance without altering the main network's CNN structure through the use of a hypernetwork. We prove that these generated filters grant the equivariance property to a regular CNN main network. Empirically, HE-CNN outperforms $G$-CNNs and achieves comparable performance to advanced state-of-the-art $G$-CNN-based methods on both types of datasets, with and without rotation augmentation.

## 1 Introduction

Convolutional neural networks (CNNs) are effective and prevalent tools for computer vision tasks. When a specific area of an image is translated to a different location, the convolution operation inherently makes the extracted feature translate similarly. This ability to comprehend and preserve translation is known as translation equivariance, and general equivariance is highly significant in deep learning. Wood & Shawe-Taylor (1996) emphasize that it is a central problem to design neural networks that exhibit invariance or equivariance to representations in machine learning.

Group Equivariant Convolutional Neural Network ($G$-CNN), introduced by Cohen & Welling (2016), is one of the most popular equivariant CNN structure. The convolution operation is modified to equivariant to rotations, translations, and reflections. The equivariance property of $G$-CNN enables effective handling of data with symmetries.

Despite the advantages of equivariance, the trade-off of $G$-CNN is that it may underperform on datasets with minimal rotational variation. To demonstrate this trade-off, we train a small CNN (3k parameters) and a $G$-CNN (12.5k parameters) on two datasets: the original MNIST dataset (Deng, 2012) and a variant of MNIST augmented by random rotations, denoted by R-MNIST (Larochelle et al., 2007). Each model is evaluated on the test set corresponding to its training dataset, and the performances are shown in Figure 1. CNN outperforms $G$-CNN on the original MNIST, while $G$-CNN outperforms CNN on the R-MNIST. Additionally, we observe that the performance of $G$-CNN on the R-MNIST is much lower than CNN on the original MNIST.[1] This imbalance of performance across different datasets can limit the expressive power and generalization capability of $G$-CNN.

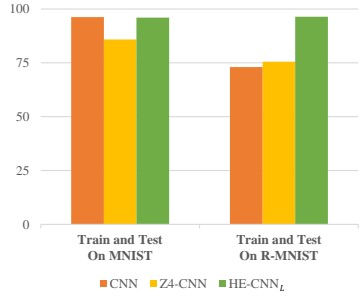

Figure 1: Performance(%) of CNN, $G$-CNN and HE-CNN$_L$ trained and tested on different MNIST variations.

---

[1]Experimental details and more comprehensive comparisons are presented in Section 5.

To achieve equivariance without the discussed trade-offs, a novel alternative is needed that does not rely on $G$-CNN filters, where achieving equivariance depends on using all rotated versions of the same filter. Such alternative is possible when the parameters of the convolutional layers are dependent on the inputs: if a single convolutional filter can inherently rotate as the input rotates, the extracted feature will be equivariant to these rotations. Therefore, we propose a novel Hypernetwork-based Equivariant CNN (HE-CNN) to achieve equivariance, generating filters that rotate as inputs rotate. Specifically, HE-CNN consists of a dynamic hypernetwork and a main network. The main network is chosen to be a general CNN. The dynamic hypernetwork generates input-dependent parameters for the main network, composed of two components: a non-equivariant parameter pieces (NEP) generator and a novel module named equivariant combiner. The NEP generator generates parameter pieces of entire parameters. The equivariant combiner, abbreviated as equi-combiner, combines the parameter pieces by averaging the output of all rotated versions of inputs to form full parameters with the ability to follow rotations on the inputs.

Theoretically, we demonstrate that the proposed HE-CNN confers the equivariance property to non-equivariant CNN main network. We also provide a light version, denoted by HE-CNN$_L$, for efficient implementation. Empirically, we show such equivariance without $G$-CNN filters enables HE-CNN to outperform base $G$-CNNs on datasets with and without rotation augmentation, while achieving performance comparable to advanced state-of-the-art $G$-CNN-based methods. As shown in Figure 1, a small HE-CNN with 11.5k parameters achieves comparable performance to the CNN on the original MNIST, while maintaining a high accuracy on the R-MNIST dataset.

Our main contribution can be summarized as follows.

1. Theoretically, we propose an alternative way to achieve equivariance: instead of constraining the filters, equivariance is achieved through input-dependent parameters.

2. We propose the HE-CNN to achieve the equivariance via a hypernetwork, and provide a light version HE-CNN$_L$ for more efficient implementation.

3. Extensive experiments demonstrate the effectiveness of the proposed HE-CNN over base $G$-CNN.

## 2 RELATED WORK

$G$-CNNs (Cohen & Welling, 2016) are introduced as one of the earliest adaptation of general equivariance into CNN, and have been the backbone of many equivariant image processing neural networks, including steerable CNNs (Cohen & Welling, 2017; Weiler et al., 2018) and spherical CNNs (Cohen et al., 2018; Salihu et al., 2024), with several applications such as domain adaptation (Zhang et al., 2022), pose estimation (Howell et al., 2023; Li et al., 2021), and so on.

One approach to make non-equivariant model equivariant is the canonicalization method (Kaba et al., 2023; Mondal et al., 2023). The equivariance in the canonicalization method is achieved through a $G$-CNN based canonicalization network, learning to rotate the input before feeding into the frozen pretrained model.

Frame-averaging (Puny et al., 2021) approximates equivariance in models by averaging non-equivariant models over sub-frames of a group. Partial Equivariance (Romero & Lohit, 2022) learns beneficial symmetries from the data distribution and retains only these symmetries in $G$-CNN networks. In exchange for improved efficiency, both methods lose strict equivariance.

*Hypernetwork*, initially introduced by Ha et al. (2016), provides an alternative approach to train a neural network. It has been used in federated learning (Shamsian et al., 2021), few-shot learning (Sendera et al., 2023; Yin et al., 2022), continual learning (Hemati et al., 2023) and so on, due to its versatility and parameter-efficiency. The network designated for training is known as the *main* network. The network responsible for generating the parameters of the main network is referred to as the *hypernetwork*. If the hypernetwork generates input-dependent parameters for the main network, we call it a *dynamic* hypernetwork, and otherwise it is *static*. The combination of a main network and a hypernetwork is referred to as a *full network* in this paper.

For hypernetworks related to equivariance, Garrido et al. (2023) use a hypernetwork for parameter sharing in equivariant models, not for achieving equivariance. To the best of our knowledge, there is no work using hypernetworks to achieve the equivariance on regular CNNs.

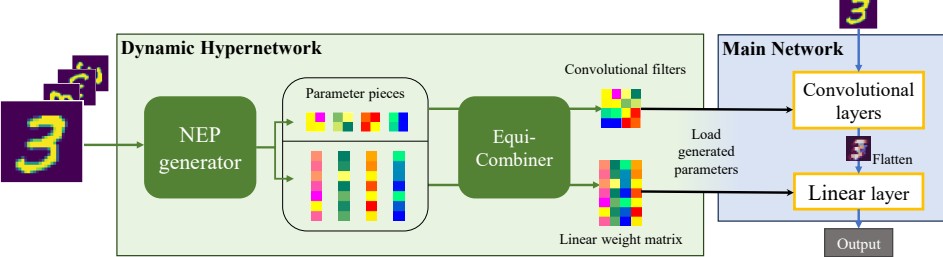

Figure 2: An overview of the HE-CNN architecture. All learnable parameters lie in the NEP generator.

## 3 PRELIMINARY

In this section, we give a brief definition of groups and representations, and a comprehensive definition can be found in Section A.1. For a set $G$ with a operation $*$, $(G, *)$ or simply $G$ is a *group* if all following properties are satisfied: $*$ is associative, $G$ has an identity element, and every element has an inverse. Given $G$ and a vector space $V$, a *representation* $\rho$ on $V$ is a mapping on $G$, where every $\rho(g)$ is a linear map on $V$ for $g \in G$. Furthermore, $\rho(g * g') = \rho(g) \circ \rho(g')$ for any $g, g' \in G$, where $\circ$ denotes function composition. A representation $\rho$ is called a *trivial* representation if for any $g \in G$, $\rho(g)x = x$ for all $x \in V$. If $\rho$ is fixed and clear from the context, $\rho(g)x$ can be simply shorten to $gx$.

The group $G$ and representation are fixed when designing a neural network based on the desired symmetries for given tasks. Now we can define equivariance and invariance in deep neural networks.

**Definition 1.** Let $f$ denote a deep neural network or a single layer in it. For two representations $\rho_1$ and $\rho_2$, we say $f$ is *(G-)equivariant* if for any input $x$, we have $\rho_2(g)f(x) = f(\rho_1(g)x)$ for any $g \in G$. If $\rho_2$ is the trivial representation, then we say $f$ is *(G-)invariant*. That is, $f(x) = f(\rho_1(g)x)$.

## 4 METHODOLOGY

Prior to the introduction of the proposed HE-CNN, we introduce the notation used in the proposed HE-CNN and adapt the definition of equivariance in this setting. When utilizing hypernetwork on a main neural network $f$, the input space is denoted by $X$, and the weight space in neural networks is denoted by $\Omega$. We denote a dynamic hypernetwork by $w : X \to \Omega$. For any input $x \in X$, the corresponding generated weight is $w(x)$, and $f_{w(x)}$ refers to the main network that loads $w(x)$ as its parameters. The full network is denoted by $f_{w(\cdot)}(\cdot)$, or simply $f_w$, which maps input $x$ to $f_{w(x)}(x)$. Within the context of dynamic hypernetworks, $f_{w(\cdot)}(\cdot)$ is $G$-equivariant if $\rho_2(g)f_{w(x)}(x) = f_{w(\rho_1(g)x)}(\rho_1(g)x)$, and invariant if $\rho_2$ is the trivial representation. With the above notations, in the following sections, we present the proposed HE-CNN model.

### 4.1 THE ARCHITECTURE

We assume input images are square-shaped.[2] As illustrated in Figure 2, the proposed HE-CNN consists of a main network $f$ and a dynamic hypernetwork $w$. The main network $f$ of HE-CNN consists of several convolutional layers, a single flatten layer, several linear layers, and some activation/pooling/batch-normalization layers in between. As the main network CNN is already equivariant to translation, we only need rotational and reflectional equivariance. We first fix the group $G$ to be $(Z_4, +_{mod\ 4})$, the group of 90-degree rotations. More general group of rotations, reflections are discussed in Section 4.4. We aim to achieve equivariance on the full network $f_w$, combining $f$ with the input-dependent parameters generated by a dynamic hypernetwork $w$. We summarize our objectives in the following definition.

**Definition 2.** A neural network $f$ is *w-based equivariant* or *hypernetwork-based equivariant* if the following conditions hold:

1. For general parameters $\gamma$, $f_\gamma$ is not equivariant.

---

[2]Due to their correspondence with rotations, input images of equivariant structures are typically selected to be square-shaped (Cohen & Welling, 2016).

2. When using $w$ to generate input-dependent parameters, $f_{w(\cdot)}(\cdot)$ is equivariant to $G$.

In other words, $f$ is $w$-based equivariant if $w$ grants equivariance to non-equivariant main network. The output of $w$ is referred to as the equivariant parameters.

The objective of the proposed HE-CNN is to ensure that $w$ generates equivariant parameters. To achieve that, $w$ consists of two components. The first component is a *non-equivariant parameter pieces generator (NEP generator)*. The NEP generator is responsible of generating approximately $1/4$ of the total parameters, including filters in convolutional layers and weight matrices for linear layers. The details for each case are introduced in the next two sections. The NEP generator can be any non-equivariant neural network, and our implementation uses regular CNNs. Additionally, we expect it to be non-equivariant to avoid collapsing to invariant filters as demonstrated in Section D.1.

When given an input image, we collect the four 90°-rotated versions of it and send them to the NEP generator, resulting in four *parameter pieces*. Then, the four parameter pieces are fed into the second component, the *equivariant combiner (equi-combiner)*, to assemble parameter pieces to get full parameters that could achieve the equivariance. The design of the equi-combiner is outlined in the following two sections, which correspond to two cases: convolutional layers and linear layers.

Recall that most $G$-CNN designs (Cohen & Welling, 2016; Worrall et al., 2017; Wang et al., 2022) extracts equivariant features in the convolutional layers and aiming for invariance in the final classification. We adapt the same design goal in HE-CNN: For the convolutional layers, the objective of the equi-combiner is to ensure the equivariance to rotations. For linear layers, the equi-combiner is to generate parameters so that the final output is invariant.

## 4.2 Hypernetwork for Convolutional Layers

Without loss of generality, in the following, we present the design of the hypernetwork that generates parameters for each convolutional layer $f$.

The group $Z_4$ has four elements: 0, 1, 2, and 3, each corresponding to 0°, 90°, 180° and 270°. The representation $\rho(g)$ is to perform counter-clockwise rotations corresponding to $g$ on input image $x$. For all the convolutional layers, we let both $\rho_1$ and $\rho_2$ in the definition of equivariance to be the same $\rho$. The NEP generator is denoted by $N$.

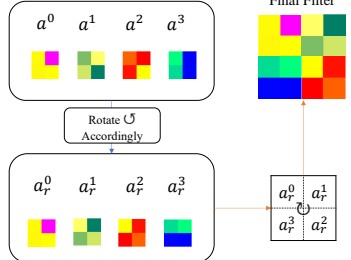

Figure 3: A visualization of the equi-combiner for a 4×4 filter.

For the convolutional layer in $f$ with $C_{\text{in}}$ input channels, $C_{\text{out}}$ output channels, and the filter dimension $K$, the weight of the convolution filter is in a shape of $(C_{\text{out}}, C_{\text{in}}, K, K)$ and the bias is in shape $(C_{\text{out}})$. The output shape of the NEP generator is $(C_{\text{out}}, C_{\text{in}}, \lceil K/2 \rceil, \lceil K/2 \rceil) + (C_{\text{out}})$, where the first part is for parameter pieces of convolution filters, the second part is for the bias, and $\lceil \cdot \rceil$ denotes the ceiling function. Accordingly, the NEP generator is denoted by $N_{\text{filters}}(x)$ for parameter pieces of convolutional filters and $N_{\text{bias}}(x)$ for biases.

We first study how to assemble full filters using $N_{\text{filters}}$. We denote the input by $x^0$, and denote its 90°, 180° and 270° rotations as $x^1, x^2$ and $x^3$ accordingly. Each of them is sent into $N_{\text{filters}}$ and the corresponding outputs (i.e., parameter pieces) are denoted by $a^i = N_{\text{filters}}(x^i)$ ($i = 0, 1, 2, 3$). The equi-combiner takes each $a^i$, and perform a **counter-clockwise** $i \cdot 90°$ rotation for each and get $a_r^i$. As illustrated in Figure 3, we assemble them in a **clockwise** manner (i.e., $a_r^0$ is on the top left, $a_r^1$ is on the top right, $a_r^2$ is on the bottom right, and $a_r^3$ is on the bottom left) to be the final filter.

For the case that $K$ is odd, we follow the same process to get the final filter with size $(K + 1, K + 1)$. Then, we average the middle ($(K+1)/2$-th and $(K+3)/2$-th) columns and rows into one. Mathematically, the above generation process of weights in a filter via the NEP generator can be formulated as

$$w_{\text{filters}}(x) = \sum_{g \in Z_4} g^{-1} E\big(N_{\text{filters}}(gx)\big),$$

where $E$ is the operation that places parameter pieces to the top left corner of a filter.

For the bias, we expect it to be identical for all the rotated versions of the input. To achieve that, we simply average the outputs of $\{N_{\text{bias}}(x^i)\}_{i=0}^3$ as

$$w_{\text{bias}}(x) = \frac{1}{4} \sum_{g \in Z_4} N_{\text{bias}}(gx). \tag{1}$$

For the hypernetwork designed above for convolutional layers, we show that it achieves the equivariance on the full network in the following theorem.[3]

**Theorem 1.** *Let $f^{conv}$ be a main neural network with several convolutional layers, and $w$ be the hypernetwork composed of the NEP generator and the equi-combiner as designed above. Then, for all $g \in Z_4$, we have*

$$g f_{w(x)}^{conv}(x) = f_{w(gx)}^{conv}(gx).$$

For an input $x$, after all convolutional layers of the full network $f_w$, we have an extracted features $\{B_i\}_{i=1}^{C_{\text{out}}} = f_{w(x)}^{conv}(x)$. For simplicity, we study the case where $C_{\text{out}} = 1$, and consequently $B = f_{w(x)}^{conv}(x)$. Prior to the next step in HE-CNN, $B$ is flattened before sending into the following linear layers of the full network. We denote the flatten operation by $P$, and the flattened vector is denoted by $v = P(B)$. If the dimension of $B$ is $d \times d$, the length of $v$ is $d^2$. For the general case where $C_{\text{out}} > 1$, we generate linear weights for all channels with the same process, and concatenate them to form the final linear weights.

## 4.3 Hypernetwork for Linear Layers

In this section, we first study the case when the main network $f$ consists of only one linear layer. The linear layer takes flattened vector $v$ as the input, and output a vector of length $m$. In this case, the weight matrix $W$ of the main network $f$ is in shape $(m, d^2)$. The hypernetwork $w$ aims to generate input-dependent $W$ so that the output of $f_w$ is invariant, i.e., $f_w(P(B)) = f_w(P(\rho(g)B))$.

We adapt the rotation representation $\rho$ for the square matrix in Section 4.2 to the flattened vectors $v$ using $\rho_{\text{lin}} = P\rho P^{-1}$, where $P^{-1}$ denotes the inverse of the flatten operation. The invariance property of $f_w$ to be achieved can be rewritten as $f_w(v) = f_w(\rho_{\text{lin}}(g)v)$.

Based on the following proposition, we can view $\rho_{\text{lin}}$ as **a collection of length-4 permutations**, which is illustrated on the top of Figure 4.

**Proposition 2.** *For the representation $\rho_{lin}$ on the vector $v$, the following statements hold:*

1. *Let $v' = \rho_{lin}(1)v$ be $\rho_{lin}$ applied to $v$ once For any entry $v_i$ of the vector $v$, there is a unique permutation $\sigma = (i, j, k, l)$ such that $v_i = v'_j$, $v_j = v'_k$, $v_k = v'_l$, and $v_l = v'_i$.[4]*

2. *The representation $\rho_{lin}$ can be viewed as a collection of all such length-4 permutations on the indices of $v$.*

Since $v$ is of length $d^2$, there are $\lceil d^2/4 \rceil$ different length-4 permutations. The collection of all such permutations is denoted by $S$. For any $\sigma = (i, j, k, l) \in S$, we denote the vector obtained after permuting the indices of a vector $v$ by $\sigma(v)$ and the matrix obtained after permuting the column vectors of a matrix $W$ by $\sigma(W)$. Then we have the following result.

**Lemma 3.** *In the main network $f$ with the input $v$ of the linear layer and the corresponding weight matrix $W$, for any given permutation $\sigma$, we have*

$$\sigma(v)\sigma(W)^\top = vW^\top, \tag{2}$$

*where superscript $^\top$ denotes the transpose operation.*

---

[3]Proofs of all theorems are presented in Section D.3.

[4]A permutation $(i, j, k, l)$ is a circular expression, that is, $(i, j, k, l) = (j, k, l, i) = (k, l, i, j) = (l, i, j, k)$. Given $i$, the exact expressions of $j$, $k$, and $l$ are given in Appendix D.2. The special case of an odd $d$ is also detailed in Appendix D.2.

Lemma 3 leads to a design goal for the hypernetwork $w$ that $w$ should generate weight matrix $W$ that permutes its columns as $v$ is permuted by $\sigma$ and if the linear layer in the main network $f$ possesses a bias, $w$ should generate the same bias regardless of the permutation.

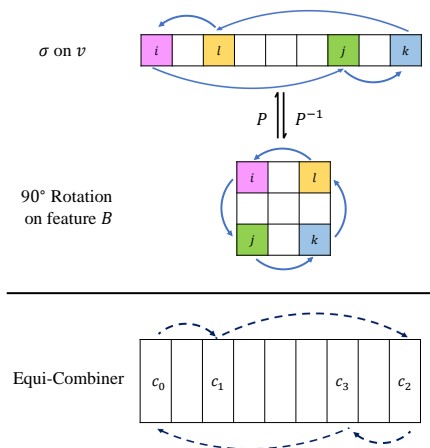

With this design goal in mind, the NEP generator $N$ generates one column vector per permutation $\sigma \in S$, and all of the bias, resulting in the output in shape $(m, \lceil d^2/4 \rceil) + (m)$. We can again split the output into $N_{\text{weight}}(x)$ and $N_{\text{bias}}(x)$ accordingly. Moreover, for each $\sigma \in S$, the corresponding generated column vector in $N_{\text{weight}}(x)$ is denoted by $N_\sigma(x)$. In the following, we fix a $\sigma = (i, j, k, l) \in S$ to better illustrate the equi-combiner. We repeat the process for all $\sigma$'s in $S$ to form the final linear weight.

Figure 4: On the top, we visualize a permutation $\sigma = (i, j, k, l)$, and the corresponding $90°$ rotation. On the bottom, the equi-combiner permutes column vectors $\{c_i\}_{i=0}^{3}$ **reverse** to $\sigma$.

Given an input image $x^0$, we again denote all rotated versions by $\{x^i\}_{0 \leq i \leq 3}$. All the four versions $\{x^i\}$ are fed into the NEP generator $N_\sigma$, and the output column vectors are denoted by $c_i = N_\sigma(x^i) \in \mathbb{R}^{m \times 1}$ for $i = 0, 1, 2, 3$. The equi-combiner places $\{c_i\}_{i=0}^{3}$ into the parameter matrix $W$ with a direction **reverse** to $\sigma = (i, j, k, l)$ as

$$W_i = c_0, W_l = c_1, W_k = c_2, W_j = c_3, \tag{3}$$

where $W_n$ denotes the $n$-th column of $W$. The bottom of Figure 4 gives an illustration of such process. Given four column vectors $\{c_i\}_{i=0}^{3}$ and the corresponding length-4 permutation $\sigma = (i, j, k, l)$, Eq. (3) defines an operation $E_{\sigma^{-1}}$ by placing each column vector $c_i$ into the corresponding column of weight matrix $W$. The generation in the hypernetwork can be formulated as

$$w_{\text{weight}}(x) = \sum_{\sigma \in S} E_{\sigma^{-1}}\left(\{N_\sigma(gx)\}_{g \in Z_4}\right).$$

Similar to the convolutional case, we expect the bias to be the same for all inputs and so we collect the output of the NEP generator and simply take the average as in Eq. (1).

For the above design, we prove the invariance property for the linear layer in the full network $f_w$ in the following theorem.

**Theorem 4.** *Let $f^{lin}$ be one single linear layer, and $w$ be the hypernetwork defined above. Then, $w$ generates equivariant parameters that grants the invariance for $f^{lin}$. That is, for all input $x$ and $g \in Z_4$, we have $f^{lin}_{w(x)}(x) = f^{lin}_{w(gx)}(gx)$.*

According to Theorem 4, we show that the proposed HE-CNN could generate outputs invariant to permutations after the first linear layer. For the rest of the linear layers (if any), all weights should be the same regardless of the permutation to preserve the invariance of the full network $f_w$. Therefore, for any additional layers, the NEP generator outputs weight $W$ for any of the rotated inputs $\{x^i\}$, and then simply average all outputs to get the weight matrix.

## 4.4 EXTENSION TO GROUP $D_{4n} \times \mathbb{R}^2$

The proposed HE-CNN can achieve $Z_4$-equivariance and CNN structure is already equivariant to translation group $\mathbb{R}^2$. In this section, we extend to $D_{4n}$, the group of reflections and $90/n^\circ$ rotations. As discussed in Section 4 of Basu et al. (2023), given an arbitrary group $G$ and a non-equivariant neural network $H$, $F(x) = \sum_{g \in G} \rho_g^{-1} H(\rho_g(x))$ is always equivariant regarding $G$. Based on this result, we choose $G$ to be the quotient group $D_{4n}/Z_4$. As the proposed $f_{w(\cdot)}$ in previous sections is already equivariant to $Z_4$ and translation, it is easy to see that $F_{w(x)}(x) = \frac{1}{n} \sum_{g \in D_{4n}/Z_4} g^{-1} f_{w(gx)}(gx)$ is equivariant to $D_{4n} \times \mathbb{R}^2$ and it is named $D_{4n}$-HE-CNN.

Table 1: Testing accuracy (%) on various MNIST datasets.

| | Train on MNIST | | | Train on R-MNIST | | |
|---|---|---|---|---|---|---|
| | CNN | $Z_4$-CNN | HE-CNN$_L$ | CNN | $Z_4$-CNN | HE-CNN$_L$ |
| MNIST | **96.29** | 85.88 | 96.03 | 75.27 | 76.22 | **96.44** |
| R-MNIST | 34.81 | 50.02 | **76.94** | 73.08 | 75.51 | **96.32** |
| 90°-MNIST | 17.76 | 85.88 | **96.03** | 75.68 | 76.22 | **96.44** |
| Average Performance | 49.62 | 73.93 | **89.67** | 74.68 | 75.98 | **96.40** |
| Equivariance difference | 68.86 | 0.00 | 0.00 | 0.27 | 0.00 | 0.00 |
| # Parameter | 3370 | 12522 | 11527 | 3370 | 12522 | 11527 |

## 4.5 LIGHT VERSION OF HE-CNNS

As discussed in Sec. 4.2 and 4.3, the NEP generator outputs a vector $v \in \mathbb{R}^l$, where $l$ is approximately equal to a quarter of the total parameters in the main network. When the main network has a large number of parameters, HE-CNNs could need a large NEP generator.

To reduce the size of the NEP generator, we first split $v$ into $\{v_i\}_{i=1}^T$, where $v_i \in \mathbb{R}^{l_i}$ contains parameters for the $i$-th layer of the main network, and $T$ is the total number of layers of the main network. For the final linear decoder $L$ in the hypernetwork $w$, we can replacing $L$ with $T$ smaller linear layers $\{L_i\}_{i=1}^T$, each generating $v_i$. Next, we replace each $L_i$ with two smaller linear decoders that generate intermediate matrices $a_i \in \mathbb{R}^{\sqrt{l_i} \times r}$ and $b_i \in \mathbb{R}^{r \times \sqrt{l_i}}$,[5] where $r \ll \sqrt{l_i}$ is a fixed rank. The target $v_i$ is obtained by flattening the product of $a_i$ and $b_i$. This approach could reduce the parameter size of each $L_i$ from $\mathcal{O}(l_i)$ to $\mathcal{O}(\sqrt{l_i})$. This light version of HE-CNN is denoted by HE-CNN$_L$, and an illustration is provided in Appendix C.1.

## 5 EXPERIMENT

In this section, we empirically evaluate HE-CNN. Models and datasets are outlined for each experiment, with additional model and hyperparameter details specified in the Appendix C.

### 5.1 TRADE-OFF OF THE EQUIVARIANCE IN G-CNN

**Models.** To effective demonstrating HE-CNN solves the imbalanced performance of $G$-CNN, models in this section are designed to be small. Specifically, (i) CNN is composed of three hidden convolutional layers, followed with one linear classification layer. (ii) G-CNN has the same hidden layers with $G = Z_4$. (iii) HE-CNN's main network is the same with CNN mentioned above. The NEP generator has 2 convolutional layers, and the light version is used to match the parameter count.

**Datasets.** We conduct experiments on various MNIST datasets, including: (i) MNIST (Deng, 2012), containing 70,000 handwritten digits. (ii) R-MNIST (Larochelle et al., 2007), a variant of the MNIST with random continuous rotations. (iii) 90°-MNIST, which rotates every sample from MNIST counter-clockwise by 90 degrees. Each dataset is split into a training set of 60,000 samples and a test set of 10,000 samples. We train models separately on either MNIST or R-MNIST and evaluate their performance across all three datasets. The average accuracy on all datasets is used to demonstrate consistency across regular and rotated data.

**Results.** Table 1 shows the testing accuracy for different models. As can be seen, HE-CNN$_L$ achieves the best accuracy in almost all settings, with the only exception when trained and tested on MNIST, performing comparably (96.03%) to CNN. This empirically shows HE-CNN$_L$ achieves equivariance without the discussed trade-off. To quantify equivariance, we introduce the equivariance difference: the relative difference (difference divided by the sum) in accuracy between MNIST and 90°-MNIST. HE-CNN$_L$ shows a 0% equivariance difference, demonstrating strict equivariance.

---

[5]If $\sqrt{l_i}$ is not an integer, we use $\lceil \sqrt{l_i} \rceil$ and truncate the extra from the final generated vector.

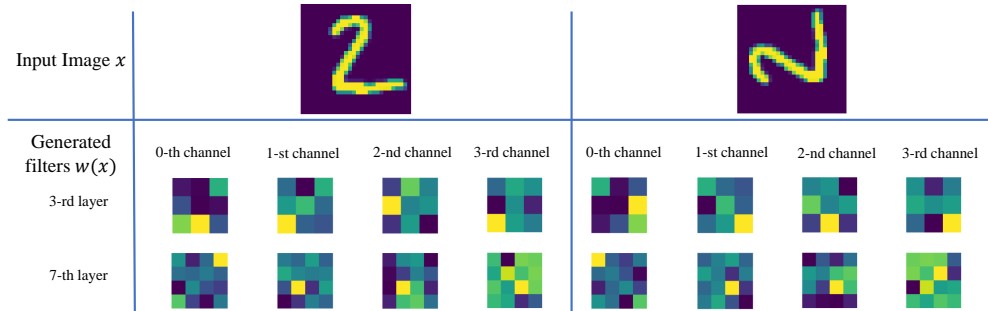

Figure 5: For two images related through 90° rotations, we visualize some of their generated filters.

## 5.2 R-MNIST

**Models.** We evaluate performance on R-MNIST using regular-sized models to assess scalability and generalization. For HE-CNN, the main CNN has seven convolutional layers, followed by two linear layers. The NEP generator is composed of three convolutional layers. We implement both regular and light versions. For G-CNN, it is set to have the same layers and channels as the main CNN of HE-CNN, and the group is chosen to be $P_4 = D_4 \times \mathbb{R}^2$. We also include state-of-the-art steerable CNNs for comparison. All models are trained on the training set and evaluated on the testing set of R-MNIST. Results are shown in Table 2.

Table 2: Classification Accuracy (%) on the R-MNIST.

| Model | Accuracy (%) |
|---|---|
| Regular CNN(Schmidt & Roth, 2014) | 96.02 |
| **Equivariance based Methods** | |
| P4-CNN (Cohen & Welling, 2016) | 97.72 |
| LieConv (Finzi et al., 2020) | 98.76 |
| Steerable-CNN (Weiler et al., 2018) | 99.27 |
| E2-CNN (Weiler & Cesa, 2019) | 99.32 |
| Sim2-CNN (Knigge et al., 2022) | 99.41 |
| **Hypernetwork based Methods** | |
| Z4-HE-CNN | **99.50** |
| Z4-HE-CNN$_L$ | 97.91 |
| D8-HE-CNN$_L$ | 98.01 |
| D16-HE-CNN$_L$ | 98.05 |

**Results.** Based on Table 2, HE-CNN outperforms previous state-of-the-art methods, while HE-CNN$_L$ ensures better performance compared to the original G-CNN method. However, we do not claim to be the new state-of-the-art method. Our model utilizes 9.68 million parameters, whereas Sim2-CNN uses only 864 thousand. It is a more fair comparison between HE-CNN$_L$ with 51.9 thousand parameters and $P_4$-CNN with 98.0 thousand parameters. With just over half the parameters of the $P_4$-CNN, HE-CNN$_L$ outperforms $P_4$-CNN while utilizing the same level of symmetry.

To verify that the generated filters rotate as the input rotates, we visualize the generated filters for a randomly selected MNIST image $x$ and its rotated version $x'$ in Figure 5. For every filter generated from $x$ (left), the corresponding filter from $x'$ (right) in the same layer and channel is precisely related by a 90-degree rotation. Numerically, we rotate the generated filters of $x'$ back and compute the MSE difference with filters generated by $x$. The value is negligibly small (less than $10^{-6}$).

## 5.3 CIFAR10/100

**Models.** Partial equivariance (Romero & Lohit, 2022), as discussed in Sec.2, learns additional information about whether certain symmetries are beneficial, keeping only the beneficial symmetries in the G-CNN filters. We use the same baseline models (Romero & Lohit, 2022), a residual network composed of two residual blocks and a 13-layer CNN (Laine & Aila, 2017). For each baseline model, we implement G-CNN and Partial Equivariance with two groups: $Z_4$ and $D_8$. For HE-CNN, we implement both the regular and light versions, with three hidden convolutional layers. Models are trained on the training set and evaluated on the test set.

**Datasets.** The CIFAR-10 and CIFAR-100 datasets (Krizhevsky, 2009) consist of 60,000 labeled 32×32 color images. CIFAR-10 includes 10 object classes, with 6,000 images per class. CIFAR-100 contains 100 fine-grained classes, each with 600 images. Both datasets follow a 5:1 training-test split.

**Results.** As demonstrated in Table 3, HE-CNN shows comparable results with partial equivariance, without the additional information required by partial equivariance. Both HE-CNN and HE-CNN$_L$ exceed the performance of $G$-CNN, in both the residual network and the deeper CNN.

Table 3: Classification Accuracy (%) on CI-FAR10/100.

| Symmetry Group | Model | CIFAR10 | CIFAR100 |
|---|---|---|---|
| $\mathbb{R}^2$ | Residual network | 83.11 | 47.99 |
| $Z4 \times \mathbb{R}^2$ | $G$-CNN | 83.73 | 52.35 |
| | Partial equivariance | **86.15** | **53.91** |
| | HE-CNN | 85.99 | 53.56 |
| | HE-CNN$_L$ | 83.92 | 52.80 |
| $D8 \times \mathbb{R}^2$ | $G$-CNN | 85.55 | 55.55 |
| | Partial Equivariance | **89.00** | **57.26** |
| | HE-CNN | 88.67 | 56.95 |
| | HE-CNN$_L$ | 86.34 | 55.73 |
| $\mathbb{R}^2$ | 13-Layer CNN | 91.21 | 67.14 |
| $Z4 \times \mathbb{R}^2$ | $G$-CNN | 89.73 | 65.97 |
| | Partial Equivariance | **92.28** | **69.83** |
| | HE-CNN | 91.95 | 66.38 |
| | HE-CNN$_L$ | 90.12 | 66.14 |
| $D8 \times \mathbb{R}^2$ | $G$-CNN | 90.55 | 67.70 |
| | Partial Equivariance | 91.99 | **70.80** |
| | HE-CNN | **92.07** | 68.89 |
| | HE-CNN$_L$ | 90.64 | 68.11 |

Table 4: Classification Accuracy (%) on STL10.

| Model | Accuracy (%) |
|---|---|
| Base WRN16/8 | 87.26 |
| **G-Equivariant Convolutions** | |
| $Z_4$-WRN16/8 | 87.89 |
| $D_8$-WRN16/8 | 88.87 |
| **E(2)-Steerable Convolutions** | |
| WRN16/8-$\{D_8 D_4 D_1\}$ | **90.20** |
| **Hypernetwork-based** | |
| $Z_4$-HE-WRN16/8 | 90.08 |
| $Z_4$-HE-WRN16/8$_L$ | 88.92 |
| $D_8$-HE-WRN16/8$_L$ | 89.13 |

## 5.4 STL10

**Models.** To process larger images, we choose the base model as the wide residual network WRN16/8 by Zagoruyko & Komodakis (2016). For comparison, we replace all convolutional layers with G-equivariant convolutions to get $Z_4$-WRN16/8 and $D_8$-WRN16/8. On STL10 classification, the current state-of-the-art equivariant model is the $E(2)$-equivariant Steerable CNN (Weiler & Cesa, 2019). Their model is denoted by WRN16/8-$\{D_8 D_4 D_1\}$, with $D_n$ denoting the group for steerable filters.

**Datasets.** The STL-10 dataset (Coates et al., 2011) is designed for unsupervised and semi-supervised learning, featuring 13,000 labeled 96×96 color images across 10 classes and 100,000 unlabeled images. Our experiment is performed on the labeled images and splits the data into an 80-20 training-testing ratio.

**Results.** As demonstrated in Table 4, HE-CNN shows comparable performance with steerable state-of-the-art models and consistently outperforms regular $G$-CNN, even with the light version.

## 5.5 ABLATION STUDY

In this section, we present an ablation study to evaluate the impact of our dynamic hypernetwork on model performance. We train a $Z_4$-CNN using a regular hypernetwork, directly generating its parameters. We refer to this model as the Dynamic $Z_4$-CNN. The results, shown in Table 5, demonstrate that directly applying a dynamic network shows no improvement in accuracy. Instead, it eliminates the equivariance property of $Z_4$-CNN. This is caused by having different sets of parameters for any input and its rotated

Table 5: Classification Accuracy tested on three different testing set. All models are trained on the R-MNIST.

| Performance (%) | $Z4$-CNN | HE-CNN | Dynamic $Z4$-CNN |
|---|---|---|---|
| MNIST | 76.22 | 96.44 | 76.19 |
| R-MNIST | 75.51 | 96.32 | 74.54 |
| 90°-MNIST | 76.22 | 96.44 | 76.03 |
| Parameter | 12522 | 11527 | 32964 |

version. Such non-equivariance hinders the effective utilization of direct dynamic hypernetworks in equivariance studies and further highlights the significance of HE-CNN in generating dynamic filters while preserving the equivariance property.

## 6 CONCLUSIONS

In this study, we propose a novel hypernetwork-based equivariant CNN as an alternative approach to equivariance.We test HE-CNN on several benchmark datasets. Comparing to $G$-CNN based state-of-the-art methods, our network showed either better or comparable results. We showed better performance in all settings compared to the base $G$-CNN.

ETHICS STATEMENT

There are no ethical concerns in this study.

REPRODUCIBILITY STATEMENT

The code is uploaded as supplementary material. Hyperparameters of all experiments are detailed in Appendix C.3.

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

APPENDIX

# A ADDITIONAL DEFINITIONS

## A.1 GROUPS AND REPRESENTATIONS

**Definition 3.** Let $G$ be a set and $*$ be an operation. Then, $(G, *)$ is a *group* if the following holds:

1. There exists a $e \in G$, such that for any $g \in G$, $g * e = e * g = g$. We call this $e$ the identity.

2. For any element $g \in G$, there exists $h \in G$ such that $g * h = h * g = e$. We call this the inverse element, and denote it as $-h$ or $h^{-1}$ depending on the context of the operation.

3. $G$ is closed under this operation. That is, for any $g_1, g_2 \in G$, $g_1 * g_2$ is always in $G$.

4. For any $g, h, k \in G$, $g * (h * k) = (g * h) * k$.

After defining a group $(G, *)$, it is common to simply refer to it as $G$ when the operation is clear from the context.

**Definition 4.** Given a group $(G, *)$, a *representation* of $G$ on a vector space $V$ is a map $\rho$ with inputs in $G$. For any $g \in G$, $\rho(g)$ is a *linear map* on $V$. Furthermore, $\rho(g_1 * g_2) = \rho(g_1) \circ \rho(g_2)$, where $\circ$ denotes function composition. We denote the representation by $(\rho, V)$ or simply $\rho$.

The definition of a group and representation might be a bit hard to understand without some background on abstract algebra. It is helpful to think of groups as a collection of symmetries, and representation as an *action* of such symmetries on a vector space. Let us go through one example, the cyclic group of order 4, $(Z_4, +_{mod\ 4})$. Readers with experience in group theory can skip the following example section.

## A.2 EXAMPLE CYCLIC GROUP: $Z_4$

The group $Z_4$ has four elements: $\{0, 1, 2, 3\}$, combined the operation of addition modulo four. Comparing the modulo addition in $Z_4$ with counter-clockwise rotations by multiples of 90 degrees, one can see some *similarity* between them. For instance, $2 + 3 = 1 \mod 4$, and a vector ends up in the same place after rotating 180 degrees and then 270 degrees, as it does after a 90-degree rotation. This *similarity* is captured by a representation. Let $V = \{(x, y) \mid x, y \in \mathbb{R}\}$, all 2D vectors. Our chosen $\rho$ maps $g \in Z_4$ to a linear map on $V$ which perform the corresponding rotation around the origin. For instance, pick $2 \in Z_4$. For any $(x, y) \in V$, $\rho(2)$ is a linear map that rotate $(x, y)$ by 180 degrees, i.e. $\rho(2)(x, y) = (-x, -y)$. Formally, $\rho(g)v = \begin{pmatrix} \cos(g\pi/2) & -\sin(g\pi/2) \\ \sin(g\pi/2) & \cos(g\pi/2) \end{pmatrix} \begin{pmatrix} x \\ y \end{pmatrix}$ and one can check that this indeed satisfies the definition of a representation.

# B ALTERNATIVE ATTEMPTS

In this section, we discuss several alternative designs for the hypernetwork and explain why we choose not to implement them in our approach.

## B.1 FORCING EQUIVARIANCE THROUGH NUMERICAL METHODS

Before deploying the equi-combiner, we tried to encourage equivariance on a full filter generated by regular hypernetworks, by simply adding another rotation loss: During training, we rotate inputs and compute their generated filters. We compute the MSE Loss between such filters and try to minimize it. Denote the hypernetwork by $w$, then we can write the rotational loss as:

$$L_{\text{rot}} = \frac{1}{4} \sum_{g \in Z_4} L_{\text{mse}}\big(w(gx), w((g+1)x)\big).$$

The performance is poor on 90 degree rotations even though the rotation loss dropped significantly. The test accuracy never surpass $50\%$ on MNIST. We hypothesize that numerical methods can

Table 6: The classification accuracy (%) of HE-CNN and HE-CNN$_N$.

|  | R-MNIST | CIFAR10 | CIFAR100 | STL10 |
|---|---|---|---|---|
| Z4-CNN | 97.72 | 83.73 | 52.35 | 87.89 |
| Z4-HE-CNN | 99.50 | **85.99** | 53.36 | **90.08** |
| Z4-HE-CNN$_N$ | **99.52** | 85.87 | **53.41** | 89.95 |
| Z4-HE-CNN$_L$ | 97.91 | 83.92 | 52.80 | 88.92 |

only achieve approximate equivariance in filters. However, this numerical approximation may be insufficient due to the inherent sensitivity of the filters. This is the reason we need strict equivariance guaranteed by our equi-combiner.

### B.2 AVERAGING THE OUTPUTS OF A GENERAL HYPERNETWORK

As the name of our Equi-Combiner suggests, our initial design is to assemble smaller parameter pieces to obtain equivariant filters. This design goal has led us to the current structure. However, it is also a naive approach to simply average the outputs of a general hypernetwork from the rotated inputs, and we denote such approach by Z4-HE-CNN$_N$.

The comparison between Z4-HE-CNN and Z4-HE-CNN$_N$ is presented in Table 6. As can be seen, the naive design did not yield better results than the current design, despite expanding the size of the generated parameters by approximately four times. Considering the extra parameters and inference time on the hypernetwork, we stick to our current design.

### B.3 STATIC LINEAR LAYERS

After the first linear layer of the main network, the extracted feature are now invariant. Therefore, HE-CNN can have static linear layers afterwards, while staying equivariant. However, optimizing static layers to function effectively with dynamic parameters may pose challenges. To avoid this potential issue and maintain a consistent network architecture, we have decided to retain the current design.

## C EXPERIMENTS DETAILS

### C.1 ADDITIONAL ILLUSTRATION

Figure 6 is an illusion of the Light Version of HE-CNNs. Instead of directly generating $l_i$ amount of parameter for the $i$-th layer, we utilize two smaller linear encoders instead.

### C.2 TRAINING PROCESS

Given a fixed main network, we first initialize the NEP generator $N$. We choose a CNN structure with output dimension described in Sections 4.2 and 4.3. During training, the original inputs are sent into the combination of NEP generator and equivariant combiner to generate equivariant parameters. The main network $f$ then loads the equivariant parameters and process the same input. We compute the chosen loss

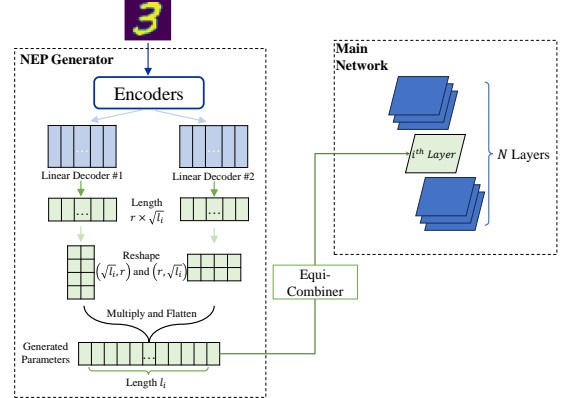

Figure 6: Visualization the light version on a single head corresponding to the $i$-th layer of the main network.

between the output of $f_w$ and the label, and preform back propagation to update the learnable parameters in the NEP generator.

When the inputs are in batch, our hypernetwork is indeed capable of generating parameters in batch. However, for inputs in batch size $b$, we would have $b$ corresponding parameters, thus $b$ main networks. This bijection can easily lead to GPU memories shortage when main network is large. We comprehend this by the following: for each batch of inputs, we average them to get one single set of parameters, used to process the batch of inputs. This significantly reduced the memory usage. However, due to the average operation, our network is equivariant if the representation $\rho(g)$ is applied on the whole batch. For each experiment, we specify whether we use parameters in batch or averaged.

### C.3 MODEL DETAIL FOR EACH EXPERIMENT

When expanding $Z_4$-HE-CNN to a larger groups $D_{4n}$, we only have to consider the extra angles in the first quadrant on the unit circle $\{90/n * i\}_{i=0}^{n-1}$. During training, we first train $f_w$ to be $Z_4$ equivariant for the first half of the training process. Then, for input $x$, we collect all input version of $x$ by angles in $\{90/n * i\}_{i=0}^{n-1}$, and sum their outputs by $f_w$ and take average as our final output.

During all experiments, we use the Adam optimizer. We noticed that it is common to observe minimal change in training loss (especially in the light version) during the first 50-100 epochs, with test accuracy typically beginning to rise after 150-250 epochs. We believe this arises from the complexities associated with learning to generate parameters. Due to the sensitivity of the generated parameters, we are very cautious about increasing the learning rate. If Adaptive average pooling layer is present in the main network and the output shape is set to 1, we modify to 2 to demonstrate our parameter generation for linear layers.

Next, we provide comprehensive details for our experiments. For the NEP generator of all cases, we provide the detail of the convolutional layers of the generator. After all convolutional layers, features are flatten and sent to a linear layer or several linear heads depending on whether light version is used.

For the first experiment where we compare CNN, $Z_4$-CNN and HE-CNN, we choose a fairly small model. Both CNN model and $Z_4$-CNN model have 3 hidden layers with 16 hidden channels and 2 by 2 filters. We use the same CNN as our main network $f$. The NEP generator is chosen to 2 hidden convolutional layers with 16 hidden channels, and the light version is utilized. We use the Adam optimizer and choose the learning rate to be 0.001 for CNN and $Z_4$-CNN, and 0.0002 for HE-CNN. Batch size is set to 32.

For the second experiment on MNIST, the main network $f$ is a CNN described in Cohen & Welling (2016). It has seven convolutional layers, six of them has 20 channels with 3 by 3 filters, and the last convolutional has 20 channels with 4 by 4 filter. Afterwards, $f$ has two linear layers with 100 intermediate channels. For the NEP generator $N$ in the hypernetwork $w$, we choose a 3-layer convolution with [16,32,32] intermediate channels, [3,3,4] filter size and [2,2,1] stride. We set the learning rate as 0.000075 and the batch size is set to 32. If the light version is utilized, our intermediate rank $r$ is set to 4.

For the experiment on CIFAR10/100, the residual main network is composed with 2 residual blocks of 32 channels, each with filter sizes 3, with additional pooling and batch normalizing layers. The main 13-layer CNN is exactly the same as in Laine & Aila (2017). In both scenarios, learning rates are set to 0.000025. Since we average batch of inputs to get one set of parameters, we choose a smaller batch size as 64 to lower the negative impact. When the light version is used, the intermediate rank $r$ is set to be 14. The NEP generator $N$ is composed of 3-layer convolution with [32,64,64] intermediate channels, with all filter sizes as 3 and stride as 2 for the last layer. ReLu, Batch normalization and Max pooling layers with filter dimension as 2 and stride as 2 are inserted between convolutional layers.

For the experiment on STL10, the main network is the wide residual network architecture. The hypernetwork generates the convolutional and linear layer, keeping the others unchanged. The learning rate is set to 0.000015 and batch size is 64. If the light version is used, the intermediate rank $r$ is chosen to be 20. The NEP generator $N$ is composed of 4-layer convolution with [64,128,128,64] intermediate channels, with all filter sizes as 3 and stride as 2 for the last layer. ReLu and Batch normalization are inserted between convolutional layers. Additionally, we perform pooling after first three convolutional layers, with kernel size as 2 and stride as 2.

# D    THEORY DETAILS AND PROOFS

## D.1    REQUIRING NON-EQUIVARIANCE IN OUR PARAMETER GENERATOR

Figure 7: Visualizing generated filters using different parameter generator. Filters on the left are ideal. On the right it collapsed to invariance, i.e., all filters are equal to their rotated versions.

The requirement of non-equivariance is we do not want relations among generated $a_r^i$. If the parameter generator is chosen to be equivariant, filters are rotational invariant as demonstrated on the right of Figure 7, which significantly reduce the expressive power.

## D.2    PERMUTATION DETAILS OF FLATTENED VECTORS

In this section, we provide the proof of Prop.2 and details of the length-4 permutations.

**Proposition 2.**  For the representation $\rho_{\text{lin}}$ on the vector $v$, the following statements hold:

1. Let $v' = \rho_{\text{lin}}(1)v$ be $\rho_{\text{lin}}$ applied on $v$ once. For any entry $v_i$ of the vector $v$, there is a unique permutation $\sigma = (i, j, k, l)$ such that $v_i = v'_j$, $v_j = v'_k$, $v_k = v'_l$, and $v_l = v'_i$.

2. The representation $\rho_{\text{lin}}$ can be viewed as a collection of all such length-4 permutations on the index of $v$.

*Proof.* $Z_4$ is cyclic (i.e. generated by one element), so we only have to show our claim holds for $\rho_{\text{lin}}(1)$. For $v_i$ as $i$-th component in $v$, denote $c = \lfloor i/d \rfloor$ and $r$ as the remainder. These two corresponds to the column and row pre-flatten accordingly. Let $j = d(d-1) - dc + r$. Then, $v'_j = v_i$. We repeat this computation on $j$ to get $k$ and $l$.

From the properties of permutations, it is well-known that $(i, j, k, l) = (j, k, l, i)$. This means the starting at $i$ or $j$ ends up with the same permutation. Therefore, we prove the uniqueness.    □

When the length of $v$ is not divisible by 4 (corresponding to an **odd** dimension $d$), the middle point $v_z$ of $v$ is unchanged. By a slight abuse of notation, we can also view the unchanged point as a permutation $(z, z, z, z)$. Based on Eq. (3), all generated column vectors $c_i$ are assigned into $W_z$. For consistency, all four column vectors are averaged into one.

This allow us to say there is $(d^2 - 1)/4 + 1$ amount of length-4 permutations. Therefore, the amount of such permutations is $d^2/4$ in general.

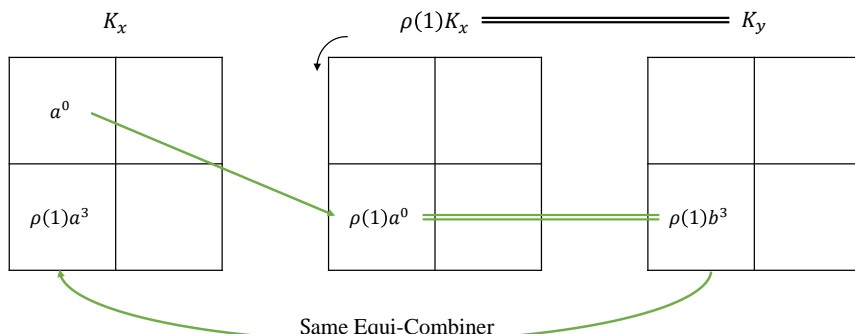

Figure 8: If the 0-th piece is assigned to the top left, the assignment of the bottom left corner is fixed.

### D.3 PROOFS OF THE MAIN THEOREMS

**Theorem 1.** Let $f^{conv}$ be a main neural network with several convolutional layers, and let $w$ be the hypernetwork composed of the NEP generator and the equi-combiner. Then, for all $g \in Z_4$, we have

$$g f^{conv}_{w(x)}(x) = f^{conv}_{w(gx)}(gx).$$

*Proof.* In the first step of the proof, we show that the designed equi-combiner offers a unique method for combining parameters that rotate as inputs rotates, up to the choice of placing the first piece $a_0$. Let $\{a_i\}_{\{0,1,2,3\}}$ be the generated parameter pieces mentioned in Section 4.2.

We can assume the dimension of target filter $d$ is even, as the odd case can be achieved by merging the middle rows and columns of the even case.

Since $Z_4$ is a cyclic group generated by 1, it suffice to prove the case of $\rho(1)$, the 90-degree rotation.

Given an input image $x^0$, let $y^0 = \rho(1)x^0 = x^1$ be the 90-degree rotated version of $x^0$. For each input $x^0$ and $y^0$, we want to generate a filter for each, denoted by $K_x$ and $K_y$. Our assumption is $K_y = \rho(1)K_x$.

For $x^0$, we denote all rotated version of input by $x^1, x^2$ and $x^3$. Similarly, we denote rotated version of $y^0$ as $y^i$ for $i = 1, 2, 3$. Since $y^0 = x^1$, we know $y^{i+1} = x^i$ for all $i = 1, 2, 3$.

Given NEP generator $N$, denoted the output $a^i = N(x^i)$ and $b^i = N(y^i)$. Due to the relationship between $x$ and $y$, we also have $b^i = a^{i+1}$.

Now for the equi-combiner $E$, recall that it assigns the parameter pieces $a^i$ and $b^i$ to the filter $K_x$ and $K_y$ accordingly, based on the index. Assume $E$ assigns the 0-th parameter pieces on the top left corner. Therefore, $a^0$ is on the top left of $K_x$. By the assumption $K_y = \rho(1)K_x$, we know that the bottom left corner is $\rho(1)a^0 = \rho(1)b^3$. Since $E$ performs the same action based on index regardless of $a$ and $b$, we know that $\rho(1)a^3$ goes to the bottom left corner as well. The process can be visualized in 8.

If we keep repeating this process, we get that $\rho(2)a^2$ goes to bottom left, $\rho(3)a^1$ goes to the top right corner. This is exactly the equi-combiner that we described, finishing the first part.

Combining the fact that the element-wise multiplication of two equal-sized square matrices remains unchanged when both matrices are subjected to 90-degree rotations, we finish the proof of the theorem.

$\square$

For linear case, we have a fairly similar proof.

**Theorem 4.** Let $f^{lin}$ be one single linear layer and $w$ be the hypernetwork in HE-CNN. Then, $w$ generate equivariant parameters that grants invariance to $f^{lin}$. That is, for all input $x$ and $g \in Z_4$, we have

$$f^{lin}_{w(x)}(x) = f^{lin}_{w(gx)}(gx).$$

*Proof.* The proof is structured into three sub-statements:

Let $\{c_i\}_{\{0,1,2,3\}}$ be the generated column vectors mentioned in Section 4.3. The following properties hold:

1. Our equi-combiner guarantees filter rotation as the inputs rotate.

2. The order of combining is unique up to the choice of placing the first piece $c_0$.

3. Let $\rho_{\text{lin}}$ be the representation in Section 4.3. If $f$ is a linear layer with input vector $v$, for any $g \in G$, we have
$$f(v) = f_{w(\rho_{\text{lin}(g)}v)}(\rho_{\text{lin}(g)}v).$$

The proof of the first and second statement is similar to the case of convolutional filters. For an input $x^0$ and a second input $y^0 = \rho(1)x^0$, we track the assignment of their generated column vectors.

The third statement directly follows from adding the same bias to both side of Eq. (2). $\square$

