# OpenReview forum: "Equivariant CNNs via Hypernetworks"
_ICLR.cc/2026/Conference — ICLR 2026 Conference Withdrawn Submission_

### Official Review · Reviewer_gubH · 2025-10-21

**Soundness:** 3
**Presentation:** 1
**Contribution:** 2
**Rating:** 4
**Confidence:** 4

**Summary:**

The paper proposes an input-dependent parameterization of convolution kernels in CNNs that ensures rotation equivariance. This is done through a hypernetwork approach where a separate network (called hypernetwork) takes the input image and outputs weights for the convolution kernels in the classification network. The weights are output in a manner which ensures rotation equivariance.

Experiments on image classification on MNIST, CIFAR10/100 and STL10 indicate that the suggested method is competitive with prior approaches for rotation equivariant networks.

**Strengths:**

The paper suggests a way to inject global image information into the weights of Equivariant CNNs. This can certainly be a fruitful direction for research. The experimental section is quite comprehensive with most standard small-scale image classification datasets included.

**Weaknesses:**

**Theory:**

In my opinion, the work would benefit from improving the connection to prior work. This would simplify the exposition and also enable further implementations for more groups. Specifically, it is well-known in the literature that in order for a convolution with kernel $K(p)$ (here $p$ is a position on the pixel grid, i.e. the variable that will be convolved over in the layer) to be equivariant with respect to trivial feature spaces on the input and output, $K$ needs to satisfy the kernel constraint $K(gp) = K(p)$ (Weiler & Cesa, 2019). With auxiliary inputs to $K$, denoted $I$, one gets $K(gp, L_g[I]) = K(p, I)$, where $L_g$ is the representation acting on $I$ (Zhdanov et al, Implicit Convolutional Kernels for Steerable CNNs, NeurIPS 2023). So $K$ can be an arbitrary invariant function of $p, I$. In the submitted paper (Section 4.2), as far as I understand, $K$ is obtained by taking $I$ to be the entire image and defining the kernel as $K(p, I) = N_{filters}(L_{g_p}[I])(g_p p)$ where $g_p$ is the rotation (a multiple of 90 degrees) such that $g_p p$ is in the upper left quadrant.

By connecting to the prior work, the generalization to $D_{4n}$ in Section 4.4 could be done without group averaging and further ways to parameterize $K$ could be explored.

Further, regarding Section 4.3, for linear layers mapping to invariant output, one gets $W(L_g[I])\rho(g) = W(I)$ which can be interpreted in terms of $W$ being equivariant wrt a particular input and output representation similar to the implicit convolutional kernels.

**Experiments:**

The computational costs (throughput) are not reported. Using a hypernetwork is a strict increase in computational cost compared to a baseline CNN, so it would be interesting to know the overhead both for the full HE-CNN and the light version.

**Questions:**

Why are there no comparisons with state of the art non-equivariant hypernetworks in the experiments?

What is the computational cost of the method?

---

### Official Review · Reviewer_39Lr · 2025-10-30

**Soundness:** 3
**Presentation:** 1
**Contribution:** 2
**Rating:** 4
**Confidence:** 4

**Summary:**

The paper proposes a novel method to build expressive equivariant models by having an equivariant hypernetwork producing input-conditional weights for a standard non-equivariant network, which performs the main inference task.
The proposed method is validated in MNIST variations, CIFAR10/100 and STL10 datasets, showing some benefits over group convolution baselines in both symmetric and natural image datasets.

**Strengths:**

As far as I know, the proposed idea is quite novel and I can see some of its potential benefits, in particular to overcome the excessive parameter sharing enforced by group-convolution designs in small architectures.
Moreover, the experimental sections shows promising results.

**Weaknesses:**

Unfortunately, I feel like the presentation quality of the current manuscript could be significantly improved.

First, I think the current introduction only provides a weak motivation, mostly based on an observation with extremely small models on MNIST.
For example, could you elaborate more about why you expect GCNNs to suffer from these trade-offs?

Second, I found the description of the proposed method to be quite unclear throughout the paper, mostly lost in the details of the instantiation of the idea on the particular architecture employed and the group $Z_4$.
Indeed, if I understood correctly, the overall idea is having a hypernetwork producing the weights $W_l$ and $b_l$ of the $l$-th linear layer s.t. $W_l(g.x) = g W_l(x) g^{-1}$ and $b_l(g.x) = g b_l(x)$.
In this way, when the $l$-th layer is applied in a deep network, all transformations commute, i.e. if $f_l$ is the concatenation of all layers $1, \dots, l$, then (ignoring non-linearities):$$f_l(g.x) =  W_l(g.x) f_{l-1}(g.x) + b_l(g.x) = g W_l(x) g^{-1} g f_{l-1}(x) + g b_l(x) =  g W_l(x) f_{l-1}(x) + g b_l(x) = g f_l(x)$$
and the overall network is equivariant. That is neat and the overall idea works regarless of how the functions $W_l(x)$ and $b_l(x)$ are implemented, which group $G$ is considered and with which representations it acts on each intermediate feature, as long as the hypernetwork has the necessary equivariance property.
Note that such formulation in terms of linear layers covers convolutional layers as well.

As a results, I feel like a lot of details in the paper can be easily moved to the appendix (e.g. details and derivations about what representation of the group Z_4 acts on the output of the flattening operation), while leaving the description in the the main paper focused on the general and, slightly more abstract, formulation.

Additionally, this formulation can be easily used for other groups beyond Z_4 (e.g. $D_4n$, s.t. frame averaging is not necessary, as done in Sec 4.4) and other architectural choices for the hypernetwork (e.g. replacing the expensive group averaging employed to make the hypernetwork equivariant).

See also other comments and questions below.

**Questions:**

From the abstract, it seems that the problem are the filters learnt by the GCNN somehow. I agree a problem when using datasets with little symmetries is that the strong weight sharing enforced by equivariance allows for too few independent filter (this is particularly problematic for small models, but less so for sufficiently wide ones). However, the paper doesn't really discuss this aspect much anymore in the paper.


Why naming the group $Z_4$? As far as I know, most works in the literature use the cyclic group $C_4$ notation.

Lines 61-64: not really clear what the authors mean by "parameter pieces" and "parameter pieces of entire parameters"

Line 65-68: these sentences could be made a bit more clear since, initially, it sounds like taking the weights outputted by an equivariant hypernetwork is sufficient to get an equivariant architecture, while it is only later clear that joint hypernetwork + main network forms an equivariant architecture.

Why using group averaging to build the hypernetwork? Isn't it more efficient to just use a GCNN? I understand a reason to use hypernets instead of GCNNs is that GCNNs can be overconstraining, especially when cosndiering non-symmetric tasks, but this seems less problematic for the hypernet itself since its task is by construction equivariant, right? Moreover, if the hypernet employs group averaging, then, why not using group averaging for the main network in the first place?

Sec. 4.4: why applying frame averaging to make the model $D_{4n}$ equivairant? The method proposed seems suitable to build $D_{4n}$ architectures directly, no?

Sec 5.5: is that a Z_4-equivairant architecture where the weights are set by a non-equivariant hypernetwork? The section could make this a bit more clear and explicit.

---

### Official Review · Reviewer_8oQN · 2025-11-02

**Soundness:** 2
**Presentation:** 2
**Contribution:** 1
**Rating:** 2
**Confidence:** 4

**Summary:**

This paper proposes a hypernetwork-based equivariant CNN as an alternative approach to achieve equivariance to $D_4$ or $C_4$. In essence, for each input, its rotated or reflected copies are fed into a hypernetwork that produces convolutional filters, which are then combined before being passed into a CNN to ensure equivariance. Experiments are conducted to verify the claim.

**Strengths:**

1. The paper is written in a clear and easy-to-follow manner.
2. The idea of using a non-equivariant hypernetwork to produce weights, which are then assembled and passed into a standard CNN to achieve equivariance, is somewhat interesting.

**Weaknesses:**

1. The motivation of this paper is not convincing. From Figure 1, I am not persuaded that (i) when trained on the original MNIST dataset, a G-CNN would suffer such a dramatic drop in performance compared with a standard CNN (from nearly 100% accuracy to around 85%), and (ii) when trained on rotated MNIST, a G-CNN would improve accuracy only from 74% to 75%. These trends contradict what has been consistently reported in the literature, such as in General E(2)-Equivariant Steerable CNNs.
2. The methodology is embarrassingly non-generalizable. What happens when the group action is not a $90^\circ$ rotation? What about scaling? It is unclear how the proposed approach can efficiently assemble weights in more general group settings.
3. I do not understand the claimed trade-off between equivariance and accuracy. The authors still constrain the model to be equivariant, only in an alternative manner. Why would that inherently increase accuracy?
4. How does the proposed model compare with a simple multiview network (or frame averaging), where rotated copies of an input are fed into a standard CNN and the outputs are averaged? After all, the authors must generate rotated copies anyway in order to produce filters.
5. None of the experiments seem to have reported computational wall-clock time, which is an important metric for evaluating practicality.

**Questions:**

See the previous section

---

### Note · Authors · 2025-11-13

I have read and agree with the venue's withdrawal policy on behalf of myself and my co-authors.